# IF THERE IS NO UNDERFITTING, THERE IS NO COLD POSTERIOR EFFECT

## ABSTRACT

The cold posterior effect (CPE) (Wenzel et al., 2020) in Bayesian deep learning shows that, for posteriors with a temperature $T < 1$, the resulting posterior predictive could have better performances than the Bayesian posterior ($T = 1$). As the Bayesian posterior is known to be optimal under perfect model specification, many recent works have studied the presence of CPE as a model misspecification problem, arising from the prior and/or from the likelihood function. In this work, we provide a more nuanced understanding of the CPE as we show that *misspecification leads to CPE only when the resulting Bayesian posterior underfits*. In fact, we theoretically show that if there is no underfitting, there is no CPE.

## 1 INTRODUCTION

In Bayesian deep learning, the cold posterior effect (CPE) (Wenzel et al., 2020) refers to the phenomenon in which if we artificially "temper" the posterior by either $p(\boldsymbol{\theta}|D) \propto (p(D|\boldsymbol{\theta})p(\boldsymbol{\theta}))^{1/T}$ or $p(\boldsymbol{\theta}|D) \propto p(D|\boldsymbol{\theta})^{1/T}p(\boldsymbol{\theta})$ with a temperature $T < 1$, the resulting posterior enjoys better predictive performance than the standard Bayesian posterior (with $T = 1$). The discovery of the CPE has sparked debates in the community about its potential contributing factors.

If the prior and likelihood are properly specified, the Bayesian solution (i.e., $T = 1$) should be optimal (Gelman et al., 2013), assuming approximate inference is properly working. Hence, the presence of the CPE implies either the prior (Wenzel et al., 2020; Fortuin et al., 2022), the likelihood (Aitchison, 2021; Kapoor et al., 2022), or both are misspecified. This has been, so far, the main argument of many works trying to explain the CPE.

One line of research examines the impact of the prior misspecification on the CPE (Wenzel et al., 2020; Fortuin et al., 2022). The priors of modern Bayesian neural networks are often selected for tractability. Consequently, the quality of the selected priors in relation to the CPE is a natural concern. Previous research has revealed that while adjusting priors can help alleviate the CPE in certain cases, there are instances where the effect persists despite such adjustments (Fortuin et al., 2022). Some studies even show that the role of priors may not be critical (Izmailov et al., 2021). Therefore, the impact of priors on the CPE remains an open question.

Furthermore, the influence of likelihood misspecification on CPE has also been investigated (Aitchison, 2021; Noci et al., 2021; Kapoor et al., 2022; Fortuin et al., 2022), and has been identified to be particularly relevant in curated datasets (Aitchison, 2021; Kapoor et al., 2022). Several studies have proposed alternative likelihood functions to address this issue and successfully mitigate the CPE (Nabarro et al., 2022; Kapoor et al., 2022). However, the underlying relation between the likelihood and CPE remains a partially unresolved question. Notably, the CPE usually emerges when data augmentation (DA) techniques are used (Wenzel et al., 2020; Izmailov et al., 2021; Fortuin et al., 2022; Noci et al., 2021; Nabarro et al., 2022; Kapoor et al., 2022). A popular hypothesis is that using DA implies the introduction of a randomly perturbed log-likelihood, which lacks a clear interpretation as a valid likelihood function (Wenzel et al., 2020; Izmailov et al., 2021). However, Nabarro et al. (2022) demonstrates that the CPE persists even when a proper likelihood function incorporating DA is defined. Therefore, further investigation is needed to fully understand their relationship.

Other works argued that CPE could mainly be an artifact of inaccurate approximate inference methods, especially in the context of neural networks, where the posteriors are extremely high dimensional and complex (Izmailov et al., 2021). However, many of the previously mentioned works have also

found setups where the CPE either disappears or is significantly alleviated through the adoption of better priors and/or better likelihoods with approximate inference methods. In these studies, the same approximate inference methods were used to illustrate, for example, how using a standard likelihood function leads to the observation of CPE and how using an alternative likelihood function removes it (Aitchison, 2021; Noci et al., 2021; Kapoor et al., 2022). In other instances, under the same approximate inference scheme, CPE is observed when using certain types of priors but it is strongly alleviated when an alternative class of priors is utilized (Wenzel et al., 2020; Fortuin et al., 2022). Therefore, there is compelling evidence suggesting that approximate methods are not, at least, a necessary condition for the CPE.

This work theoretically and empirically shows that the presence of the CPE implies the existence of underfitting, i.e., *if there is no underfitting, there is no CPE*. This perspective, in combination with the previous evidence that *CPE implies that either the likelihood, the prior or both are misspecified* (Gelman et al., 2013), allows us to deduce that CPE implies both misspecification and underfitting. And, in consequence, the underfitting of the Bayesian posterior would be induced by the very misspecification itself. This provides a more nuance perspective about the factors contributing to CPE. This conclusion can not be directly deduced from the misspecification argument itself, because prior and/or likelihood misspecification can lead Bayesian methods to both underfitting or overfitting. Both cases have been widely discussed in the literature (Domingos, 2000; Immer et al., 2021; Kapoor et al., 2022). As a result, according to our work, mitigating CPE requires addressing both misspecification and underfitting at the same time or, equivalently, designing more *flexible* likelihood functions and/or prior distributions. Importantly, we also show that this insight directly applies to exact Bayesian inference, discarding that the connection between CPE and underfitting is simply a side-effect of the use of approximate inference methods (Wenzel et al., 2020).

**Contributions**   **(i)** We theoretically demonstrate that the presence of the CPE implies there is underfitting in Section 3. **(ii)** We show that likelihood misspecification and prior misspecification result in CPE only if they also induce underfitting in Section 4. **(iii)** We show that data augmentation results in CPE only if it also induces underfitting in Section 5. In summary, we demonstrate theoretically and empirically that the presence of the CPE implies that the Bayesian posterior is underfitting. Therefore, addressing CPE requires addressing the underfitting of the Bayesian posterior.

## 2 BACKGROUND

### 2.1 NOTATION

Let us start by introducing basic notation. Consider a supervised learning problem with the sample space $\mathcal{Y} \times \mathcal{X}$. Let $D = \{(\boldsymbol{y}_i, \boldsymbol{x}_i)\}_{i=1}^n$ denote the training data, which we assume to be generated from an unknown data-generating distribution $\nu$ on $\mathcal{Y} \times \mathcal{X}$. We also assume we have a family of probabilistic models parameterized by $\boldsymbol{\Theta}$, where each $\boldsymbol{\theta}$ defines a conditional probability distribution denoted by $p(\boldsymbol{y}|\boldsymbol{x}, \boldsymbol{\theta})$. The standard metric to measure the quality of a probabilistic model $\boldsymbol{\theta}$ on a sample $(\boldsymbol{y}, \boldsymbol{x})$ is the (negative) log-loss $-\ln p(\boldsymbol{y}|\boldsymbol{x}, \boldsymbol{\theta})$. The expected (or population) loss of a probabilistic model $\boldsymbol{\theta}$ is defined as $L(\boldsymbol{\theta}) = \mathbb{E}_{(\boldsymbol{y}, \boldsymbol{x}) \sim \nu}[-\ln p(\boldsymbol{y}|\boldsymbol{x}, \boldsymbol{\theta})]$, and the empirical loss of the model $\boldsymbol{\theta}$ on the data $D$ is defined as $\hat{L}(D, \boldsymbol{\theta}) = -\frac{1}{n} \sum_{i \in [n]} \ln p(\boldsymbol{y}_i|\boldsymbol{x}_i, \boldsymbol{\theta}) = -\frac{1}{n} \ln p(D|\boldsymbol{\theta})$. We might interchange the loss expression, $\hat{L}(D, \boldsymbol{\theta})$, and the negative log-likelihood expression, $-\frac{1}{n} \ln p(D|\boldsymbol{\theta})$, in the paper for presentation. Also, if induce no ambiguity, we use $\mathbb{E}_\nu[\cdot]$ as a shorthand for $\mathbb{E}_{(\boldsymbol{y}, \boldsymbol{x}) \sim \nu}[\cdot]$.

### 2.2 (GENERALIZED) BAYESIAN LEARNING

In Bayesian learning, we learn a probability distribution $\rho(\boldsymbol{\theta}|D)$, often called a posterior, over the parameter space $\boldsymbol{\Theta}$ from the training data $D$. Given a new input $\boldsymbol{x}$, the posterior $\rho$ makes the prediction about $\boldsymbol{y}$ through (an approximation of) *Bayesian model averaging (BMA)* $p(\boldsymbol{y}|\boldsymbol{x}, \rho) = \mathbb{E}_{\boldsymbol{\theta} \sim \rho}[p(\boldsymbol{y}|\boldsymbol{x}, \boldsymbol{\theta})]$, where the posterior $\rho$ is used to combine the predictions of the models. Again, if induce no ambiguity, we use $\mathbb{E}_\rho[\cdot]$ as a shorthand for $\mathbb{E}_{\boldsymbol{\theta} \sim \rho}[\cdot]$. The predictive performance of such BMA is usually measured by the Bayes loss, defined by

$$B(\rho) = \mathbb{E}_\nu[-\ln \mathbb{E}_\rho[p(\boldsymbol{y}|\boldsymbol{x}, \boldsymbol{\theta})]]. \tag{1}$$

For some $\lambda > 0$ and a prior $p(\boldsymbol{\theta})$, the so-called *tempered posteriors* (or the generalized Bayes posterior) (Barron and Cover, 1991; Zhang, 2006; Bissiri et al., 2016; Grünwald and van Ommen, 2017), are defined as a probability distribution

$$p^\lambda(\boldsymbol{\theta}|D) \propto p(D|\boldsymbol{\theta})^\lambda p(\boldsymbol{\theta}) \,. \tag{2}$$

Even though many works on CPE use the parameter $T = 1/\lambda$ instead, we adopt $\lambda$ in the rest of the work for the convenience of derivations. Therefore, the CPE ($T < 1$) corresponds to when $\lambda > 1$. We also note that while some works study CPE with a full-tempering posterior, where the prior is also tempered, many works also find CPE for likelihood-tempering posterior (see (Wenzel et al., 2020) and the references therein). Also, with some widely chosen priors (e.g., zero-centered Gaussian priors), the likelihood-tempering posteriors are equivalent to full-tempering posteriors with rescaled prior variances (Aitchison, 2021; Bachmann et al., 2022).

When $\lambda = 1$, the tempered posterior equals the (standard) Bayesian posterior. The tempered posterior can be obtained by optimizing a generalization of the so-called (generalized) ELBO objective (Alquier et al., 2016; Higgins et al., 2017), which, for convenience, we write as follows:

$$p^\lambda(\boldsymbol{\theta}|D) = \arg\min_\rho \mathbb{E}_\rho[-\ln p(D|\boldsymbol{\theta})] + \frac{1}{\lambda} \mathrm{KL}(\rho(\boldsymbol{\theta}|D), p(\boldsymbol{\theta})) \,. \tag{3}$$

The first term is known as the (un-normalized) *reconstruction error* or the empirical Gibbs loss of the posterior $\rho$ on the data $D$, denoted as $\hat{G}(\rho, D) = \mathbb{E}_\rho[-\frac{1}{n} \ln p(D|\boldsymbol{\theta})]$, which further equals to $\mathbb{E}_\rho[\hat{L}(D, \boldsymbol{\theta})]$. Therefore, it is often used as the *training loss* in Bayesian learning (Morningstar et al., 2022). The second term in Eq. (3) is a Kullback-Leibler divergence between the posterior $\rho(\boldsymbol{\theta}|D)$ and the prior $p(\boldsymbol{\theta})$ scaled by a hyper-parameter $\lambda$.

If it induces no ambiguity, we will use $p^\lambda$ as a shorthand for $p^\lambda(\boldsymbol{\theta}|D)$. So, for example, $B(p^\lambda)$ would refer to the expected Bayes loss of the tempered-posterior $p^\lambda(\boldsymbol{\theta}|D)$. In the rest of this work, we will interpret the CPE as how changes in the parameter $\lambda$ affect the *test error* and the *training error* of $p^\lambda$ or, equivalently, the Bayes loss $B(p^\lambda)$ and the empirical Gibbs loss $\hat{G}(p^\lambda, D)$.

## 3 THE PRESENCE OF THE CPE IMPLIES UNDERFITTING

A standard understanding for underfitting refers to a situation when the trained model is not able to properly capture the relationship between input and output in the data-generating process, and results in high errors on both the training data and testing data. In the context of highly flexible model classes, like neural networks, underfittings refers to the phenomenon of learning a model having training and testing losses (much) higher than they could be, i.e., there exists another models in the model class having simultaneously lower training and testing losses. In the context of Bayesian inference, we argue that the Bayesian posterior is underfitting if there exists another posterior distribution with lower empirical Gibbs and Bayes losses at the same time.

As previously discussed, the CPE describes the phenomenon of getting better predictive performance when we make the parameter of the tempered posterior, $\lambda$, higher than 1. The next definition introduces a formal characterization. *We do not claim this is the best possible formal characterization.* However, through the rest of the paper, we will show that this simple characterization is enough to understand the relationship between CPE and underfitting.

**Definition 1.** *We say there is a CPE for Bayes loss if and only if the gradient of the Bayes loss of the posterior $p^\lambda$, $B(p^\lambda)$, evaluated at $\lambda = 1$ is negative. That is,*

$$\nabla_\lambda B(p^\lambda)_{|\lambda=1} < 0 \,, \tag{4}$$

*where the magnitude of the gradient $\nabla_\lambda B(p^\lambda)\big|_{\lambda=1}$ defines the strength of the CPE.*

According to the above definition, a (relatively large) negative gradient $\nabla_\lambda B(p^\lambda)_{|\lambda=1}$ implies that by making $\lambda$ slightly greater than 1, we will have a (relatively large) reduction in the Bayes loss with respect to the Bayesian posterior. Note that if the gradient $\nabla_\lambda B(p^\lambda)_{|\lambda=1}$ is not relatively large and negative, then we can not expect a relatively large reduction in the Bayes loss and, in consequence, the CPE will not be significant. Obviously, this formal definition could also be extended to other

specific $\lambda$ values different from 1, or even consider some aggregation over different $\lambda > 1$ values. We will stick to this definition because it is simpler, and the insights and conclusions extracted here can be easily extrapolated to other similar definitions involving the gradient of the Bayes loss.

Next, we present another critical observation. We postpone the proofs in this section to Appendix A.

**Theorem 2.** *The gradient of the empirical Gibbs loss of the tempered posterior $p^\lambda$ satisfies*

$$\forall \lambda \geq 0 \quad \nabla_\lambda \hat{G}(p^\lambda, D) = -\mathbb{V}_{p^\lambda}\big(\ln p(D|\boldsymbol{\theta})\big) \leq 0 \,, \tag{5}$$

*where $\mathbb{V}(\cdot)$ denotes the variance.*

As shown in Proposition 5 in Appendix A, to achieve $\mathbb{V}_{p^\lambda}\big(\ln p(D|\boldsymbol{\theta})\big) = 0$, we need $p^\lambda(\boldsymbol{\theta}|D) = p(\boldsymbol{\theta})$, implying that the data has no influence on the posterior. In consequence, in practical scenarios, $\mathbb{V}_{p^\lambda}\big(\ln p(D|\boldsymbol{\theta})\big)$ will always be greater than zero. Thus, increasing $\lambda$ will monotonically reduce the empirical Gibbs loss $\hat{G}(p^\lambda, D)$ (i.e., the *train error*) of $p^\lambda$. The next result also shows that the empirical Gibbs loss of the Bayesian posterior $\hat{G}(p^{\lambda=1})$ cannot reach its *floor* to observe the CPE,

**Proposition 3.** *A necessary condition for the presence of the CPE, as defined in Definition 1, is that*

$$\hat{G}(p^{\lambda=1}, D) > \min_{\boldsymbol{\theta}} - \ln p(D|\boldsymbol{\theta}) \,.$$

**Insight 1.** *Definition 1 in combination with Theorem 2 state that if the CPE is present, by making $\lambda > 1$, the test loss $B(p^\lambda)$ and the empirical Gibbs loss $\hat{G}(p^\lambda, D)$ will be reduced at the same time. Furthermore, Proposition 3 states that the Bayesian posterior still has room to fit the training data further (e.g., by placing more probability mass on the maximum likelihood estimator). From here, we can deduce that the presence of CPE implies that the Bayesian posterior ($\lambda = 1$) is underfitting, because there exists another posterior (i.e, $\rho^\lambda(\boldsymbol{\theta}|D)$ with $\lambda > 1$) that has lower training (Proposition 3) and testing (Definition 1) loss at the same time. In short, if there is CPE, the Bayesian posterior is underfitting. Or, equivalently, if the Bayesian posterior does not underfit, there is no CPE.*

However, a final question arises: when is $\lambda = 1$ (the Bayesian posterior) *optimal*? More precisely, when does the gradient of the Bayes loss with respect to $\lambda$ evaluated at $\lambda = 1$ become zero ($\nabla_\lambda B(p^\lambda)_{|\lambda=1} = 0$)? This would imply that neither (infinitesimally) increasing nor decreasing $\lambda$ changes the predictive performance. We will see that this condition is closely related to the situation that updating the Bayesian posterior with more data does not enhance its fit to the original training data better. In other words, when the Bayesian posterior contains more information about the data-generating distribution, it continues to *fit the original training data in a similar manner*.

We start by denoting $\tilde{p}^\lambda(\boldsymbol{\theta}|D, (\boldsymbol{y}, \boldsymbol{x}))$ as the distribution obtained by updating the posterior $p^\lambda(\boldsymbol{\theta}|D)$ with one new sample $(\boldsymbol{y}, \boldsymbol{x})$, i.e., $\tilde{p}^\lambda(\boldsymbol{\theta}|D, (\boldsymbol{y}, \boldsymbol{x})) \propto p(\boldsymbol{y}|\boldsymbol{x}, \boldsymbol{\theta})p^\lambda(\boldsymbol{\theta}|D)$. And we also denote $\bar{p}^\lambda$ as the distribution resulting from averaging $\tilde{p}^\lambda(\boldsymbol{\theta}|D, (\boldsymbol{y}, \boldsymbol{x}))$ over different *unseen* samples from the data-generating distribution $(\boldsymbol{y}, \boldsymbol{x}) \sim \nu(\boldsymbol{y}, \boldsymbol{x})$:

$$\bar{p}^\lambda(\boldsymbol{\theta}|D) = \mathbb{E}_\nu \big[\tilde{p}^\lambda(\boldsymbol{\theta}|D, (\boldsymbol{y}, \boldsymbol{x}))\big] \,. \tag{6}$$

In this sense, $\bar{p}^\lambda$ represents how the posterior $p^\lambda$ would be, on average, after being updated with a new sample from the data-generating distribution. This updated posterior contains a bit more information about the data-generating distribution, compared to $p^\lambda$. Using the updated posterior $\bar{p}^\lambda$, the following result introduces a characterization of the *optimality* of the Bayesian posterior.

**Theorem 4.** *The gradient of the Bayes loss at $\lambda = 1$ is null, i.e., $\nabla_\lambda B(p^\lambda)_{|\lambda=1} = 0$, if and only if,*

$$\hat{G}(p^{\lambda=1}, D) = \hat{G}(\bar{p}^{\lambda=1}, D) \,.$$

**Insight 2.** *The Bayesian posterior is optimal if after updating it using the procedure described in Eq. (6), or in other words, after exposing the Bayesian posterior to more data from the data-generating distribution, the empirical Gibbs loss over the initial training data remains unchanged.*

## 4    PRIOR MISSPECIFICATION, LIKELIHOOD MISSPECIFICATION AND THE CPE

In light of the theoretical characterization of the CPE given above in terms of underfitting, we will revisit the main arguments by previous works in relation to CPE, and we will show how we can

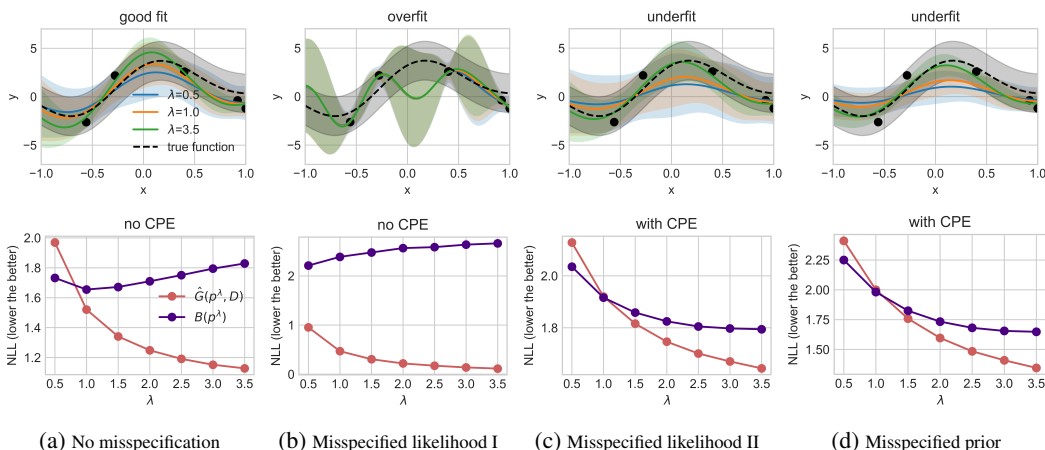

Figure 1: **1. The CPE occurs in Bayesian linear regression with exact inference. 2. Model misspecification can lead to overfitting and a "warm" posterior effect (WPE).** Every column displays a specific setting, as indicated in the caption. The first row shows exact Bayesian posterior predictive fits for three different values of the tempering parameter $\lambda$. The second row shows the Gibbs loss $\hat{G}(p^\lambda, D)$ (aka training loss) and the Bayes loss $B(p^\lambda)$ (aka testing loss) with respect to $\lambda$. The experimental details are given in Appendix C.

provide a new and more nuanced perspective on the underlying implications of the presence of the CPE. For now, we set aside data augmentation, which will be specifically treated in the next section.

**CPE, approximate inference, and NNs:** As mentioned in the introduction, several works have discussed that CPE is an artifact of inappropriate approximate inference methods, especially in the context of the highly complex posterior that emerge from neural networks (Wenzel et al., 2020). The main reasoning is that if the approximate inference method is accurate enough, the CPE disappears (Izmailov et al., 2021). However, Theorem 2 shows that when $\lambda$ is made larger than 1, the *training loss* of the exact Bayesian posterior decreases; if the *test loss* decreases too, the exact Bayesian posterior underfits. Figure 1 shows examples of a Bayesian linear regression model learned on synthetic data. Here, the exact Bayesian posterior can be computed, and it is clear from Figures 1c and 1d that the CPE can occur in Bayesian linear regression with exact inference. Although simple, the setting is articulated specifically to mimic the classification tasks using BNNs where CPE was observed. In particular, the linear model has more parameters than observations (i.e. it's overparameterized).

**Model misspecification, CPE, and underfitting:** Prior and/or likelihood misspecification can lead Bayesian methods to both underfitting or overfitting. Both cases have been widely discussed in the literature (Domingos, 2000; Immer et al., 2021; Kapoor et al., 2022). We illustrate this using a Bayesian linear regression model: Figures 1c and 1d show how the Bayesian posterior underfits due to likelihood and prior misspecification, respectively. On the other hand, Figure 1b illustrates a scenario where likelihood misspecification can perfectly lead to overfitting as well, giving rise to what we term a "warm" posterior effect (WPE), i.e., there exists other posteriors ($p^\lambda$ with $\lambda < 1$) with lower testing loss, which, at the same time, also have higher training loss due to Theorem 2.

**The prior misspecification argument:** As discussed in previous works, such as in Wenzel et al. (2020); Fortuin et al. (2022), isotropic Gaussian priors are commonly chosen in modern Bayesian neural networks for the sake of tractability in approximate Bayesian inference rather than chosen based on their alignment with our actual beliefs. Given that the presence of the CPE implies that either the likelihood and/or the prior are misspecified, and given that neural networks define highly flexible likelihood functions, there are strong reasons for thinking these commonly used priors are misspecified. Notably, the experiments conducted by Fortuin et al. (2022) demonstrate that the CPE can be mitigated in fully connected neural networks when using heavy-tailed prior distributions that better capture the weight characteristics typically observed in such networks. However, such priors were found to be ineffective in addressing the CPE in convolutional neural networks (Fortuin et al., 2022), indicating the challenges involved in designing effective Bayesian priors within this context.

Our theoretical analysis offers a more nuanced perspective on these findings: according to our analysis, if there is no underfitting, there is no CPE. Therefore, under the assumption that the

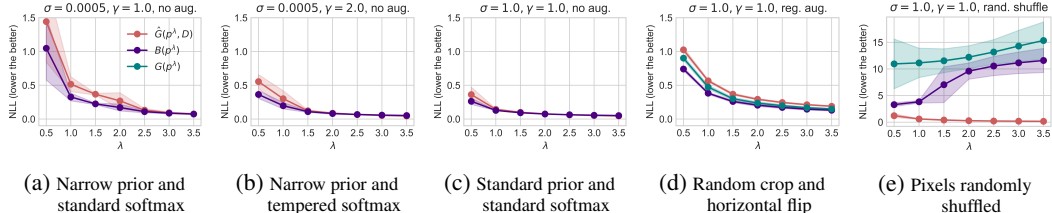

Figure 2: **Experimental illustrations for the arguments in Sections 4 and 5 using small CNN via SGLD on MNIST. We show similar results on Fashion-MNIST with small CNN and CIFAR-10(0) with ResNet-18 in Appendix C**. Figures 2a to 2c illustrate the arguments in Section 4, while Figures 2c to 2e illustrate the arguments in Section 5. Figure 2c uses the standard prior ($\sigma = 1$) and the standard softmax ($\gamma = 1$) for the likelihood without applying DA. Figure 2a follows a similar setup except for using a narrow prior. Figure 2b uses a narrow prior as in Figure 2a but with a tempered softmax that results in a lower aleatoric uncertainty. Figure 2d follows the setup as in Figure 2c but with standard DA applied, while Figure 2e uses fabricated DA. We report the training loss $\hat{G}(p^\lambda, D)$ and the testing losses $B(p^\lambda)$ and $G(p^\lambda)$ from 10 samples of the small Convolutional neural network (CNN) via Stochastic Gradient Langevin Dynamics (SGLD). We show the mean and standard error across three different seeds. For additional experimental details, please refer to Appendix C.

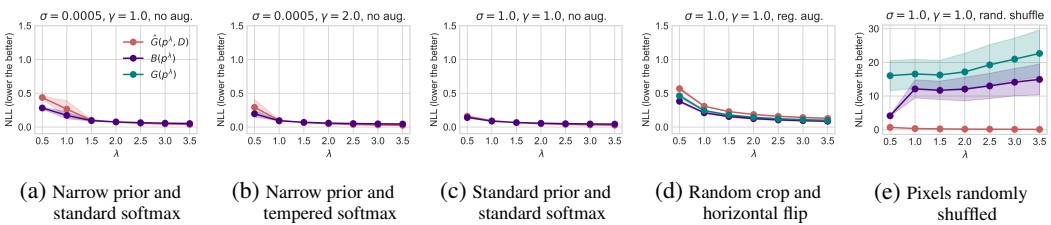

Figure 3: **Experimental illustrations for the arguments in Sections 4 and 5 using large CNN via SGLD on MNIST. We show similar results on Fashion-MNIST with large CNN and CIFAR-10(0) with ResNet-50 in Appendix C.** The experiment setup is similar to the setups in Figure 2 but with a large CNN. Please refer to Appendix C for further details on the model.

likelihood function defined by a large NN is flexible enough and is not misspecified, underfitting arises when the prior is not placing enough probability mass on models that effectively explain the training data and generalize well. Consequently, the resulting Bayesian posterior will also not place enough probability mass on those models. When this happens, our analysis shows that by increasing the parameter $\lambda$, we *discount* the impact of the prior, which reduces the "regularization effect" of the prior, alleviating the underfitting and resulting in improved training and predictive performance. In short, our analysis suggests that commonly used Bayesian priors overregularize.

Figures 2a and 2c exemplify this situation: when the prior is too narrow ($\sigma = 0.0005$) and induces a very strong regularization, the resulting posterior severely underfits the training data and leads to a high empirical Gibbs loss that deviates significantly from zero (Figure 2a). We also observe a strong CPE in this case, i.e., the Bayes loss $B(p^\lambda)$ significantly decreases when $\lambda > 1$. However, by using a flatter prior (Figure 2c) there is less underfitting, and the CPE is considerably diminished.

**The likelihood misspecification argument:** Likelihood misspecification has also been identified as a cause of CPE, especially in cases where the dataset has been *curated* (Aitchison, 2021; Kapoor et al., 2022). Data curation often involves carefully selecting samples and labels to improve the quality of the dataset. As a result, the curated data-generating distribution typically presents very low aleatoric uncertainty, meaning that $\nu(\boldsymbol{y}|\boldsymbol{x})$ usually takes values very close to either 1 or 0. However, the standard likelihoods used in deep learning, like softmax or sigmoid, implicitly assume a higher level of aleatoric uncertainty in the data (Aitchison, 2021; Kapoor et al., 2022). Therefore, their use in curated datasets, that exhibit low uncertainty, made them misspecified (Kapoor et al., 2022; Fortuin et al., 2022). To address this issue, alternative likelihood functions like the Noisy-Dirichlet model (Kapoor et al., 2022, Section 4) have been proposed, which better align with the characteristics of the curated data. On the other hand, introducing noise labels also alleviates the CPE, as demonstrated in Aitchison (2021, Figure 7). By introducing noise labels, we intentionally increase aleatoric uncertainty in the data-generating distribution, which aligns better with the high aleatoric uncertainty

assumed by the standard Bayesian deep networks (Kapoor et al., 2022). Consequently, according to these works, the CPE can be strongly alleviated when the likelihood misspecification is addressed.

Our theoretical analysis aligns with these findings. However, we can say that the key underlying cause of CPE under data curation is the underfitting induced by likelihood misspecification. The presence of underfitting is not mentioned at all by any of these previous works (Aitchison, 2021; Kapoor et al., 2022). However, it is obvious that fitting low aleatoric uncertainty data-generating distributions, e.g., $\nu(y|\boldsymbol{x}) \in \{0.01, 0.99\}$, with high aleatoric uncertainty likelihood functions e.g., $p(y|\boldsymbol{x}, \boldsymbol{\theta}) \in [0.2, 0.8]$, induces underfitting. In this case, raising the likelihood function to a power larger than 1, i.e., $p(y|\boldsymbol{x}, \boldsymbol{\theta})^\lambda$ for $\lambda > 1$, allows the likelihood to capture the low-level aleatoric uncertainty distributions better, which helps alleviate underfitting and CPE.

Figures 2a and 2b illustrate this point. Specifically, Figure 2b depicts a binary classification situation, using the same narrow prior as in Figure 2a. However, it uses a tempered softmax model defined by $p(y|\boldsymbol{x}, \boldsymbol{\theta}) = (1 + \exp(-\gamma \operatorname{logits}(\boldsymbol{x}, \boldsymbol{\theta})))^{-1}$ with $\gamma = 2$. In this case, the tempered softmax model can better represent distributions with lower aleatoric uncertainty compared to when $\gamma = 1$, corresponding to the likelihood function used in Figure 2a. Therefore, Figure 2b has a milder CPE compared with Figure 2a. However, it is important to note that the interplay between the likelihood and the prior cannot be ignored.

**Model size, CPE, and underfitting:** Larger models have the capacity to fit data more effectively, while smaller models are more likely to underfit. As we have argued that if there is no underfitting, there is no CPE, we expect that the size of the model has an impact on the strength of CPE as well, as demonstrated in Figure 2 and Figure 3. Specifically, in our experiments presented in Figure 2, we utilize a relatively small convolutional neural network (CNN), which has a more pronounced underfitting behavior, and this indeed corresponds to a stronger CPE. On the other hand, we employ a larger CNN in Figure 3, which has less underfitting, and we see the CPE is strongly alleviated.

## 5 DATA AUGMENTATION (DA) AND THE CPE

Machine learning is applied to many different fields and problems, and in many of them, the data-generating distribution is known to have properties that can be exploited to artificially generate new data samples (Shorten and Khoshgoftaar, 2019). This is commonly known as *data augmentation (DA)* and relies on the property that for a given set of transformations $T$, the data-generating distribution satisfies $\nu(\boldsymbol{y}|\boldsymbol{x}) = \nu(\boldsymbol{y}|t(\boldsymbol{x}))$ for all $t \in T$. In practice, not all the transformations are applied to every single data. Instead, a probability distribution (usually uniform) $\mu_T$ is defined over $T$, and augmented samples are drawn accordingly. As argued in Nabarro et al. (2022), the use of data augmentation when training Bayesian neural networks implicitly targets the following (pseudo) log-likelihood, denoted $\hat{L}_{\mathrm{DA}}(D, \boldsymbol{\theta})$ and defined as

$$\hat{L}_{\mathrm{DA}}(D, \boldsymbol{\theta}) = \frac{1}{n} \sum_{i \in [n]} \mathbb{E}_{t \sim \mu_T} \left[ -\ln p(\boldsymbol{y}_i | t(\boldsymbol{x}_i), \boldsymbol{\theta}) \right] , \tag{7}$$

where data augmentation provides unbiased estimates of the expectation under the set of transformations using *Monte Carlo samples* (i.e., random data augmentations).

Although some argue that this data-augmented *(pseudo) log-likelihood* "does not have a clean interpretation as a valid likelihood function" (Wenzel et al., 2020; Izmailov et al., 2021), we do not need to enter into this discussion to understand why the CPE emerges when using the generalized Bayes posterior (Bissiri et al., 2016) associated to this *(pseudo) log-likelihood*, which is the main goal of this section. We call this posterior the DA-tempered posterior and is denoted by $p_{\mathrm{DA}}^\lambda(\boldsymbol{\theta}|D)$. The DA-tempered posterior can be expressed as the global minimizer of the following learning objective,

$$p_{\mathrm{DA}}^\lambda(\boldsymbol{\theta}|D) = \arg \min_\rho \mathbb{E}_\rho[n\hat{L}_{\mathrm{DA}}(D, \boldsymbol{\theta})] + \frac{1}{\lambda} \operatorname{KL}(\rho(\boldsymbol{\theta}|D), p(\boldsymbol{\theta})) . \tag{8}$$

This is similar to Eq. (3) but now using $\hat{L}_{\mathrm{DA}}(D, \boldsymbol{\theta})$ instead of $\hat{L}(D, \boldsymbol{\theta})$, where we recall the notation $\hat{L}(D, \boldsymbol{\theta}) = -\frac{1}{n} \ln p(D|\boldsymbol{\theta})$. Hence, the resulting DA-tempered posterior is given by $p_{\mathrm{DA}}^\lambda(\boldsymbol{\theta}|D) \propto e^{-n\lambda \hat{L}_{\mathrm{DA}}(D, \boldsymbol{\theta})} p(\boldsymbol{\theta})$. In comparison, the tempered posterior $p^\lambda(\boldsymbol{\theta}|D)$ in Eq. (2) can be similarly expressed as $e^{-n\lambda \hat{L}(D, \boldsymbol{\theta})} p(\boldsymbol{\theta})$.

There is large empirical evidence that DA induces a stronger CPE (Wenzel et al., 2020; Izmailov et al., 2021; Fortuin et al., 2022). Indeed, many of these studies show that if CPE is not present in our Bayesian learning settings, using DA makes it appear. According to our previous analysis, this means that the use of DA induces underfitting. To understand why this is case, we will take a step back and begin analyzing the impact of DA in the so-called Gibbs loss of the DA-Bayesian posterior $p_{\text{DA}}^{\lambda=1}$ rather than the Bayes loss, as this will help us in understanding this puzzling phenomenon.

## 5.1 DATA AUGMENTATION AND CPE ON THE GIBBS LOSS

The expected Gibbs loss of a given posterior $\rho$, denoted $G(\rho)$, is a commonly used metric in the theoretical analysis of the *generalization performance* of Bayesian methods (Germain et al., 2016; Masegosa, 2020). The Gibbs loss represents the average of the expected log-loss of individual models under the posterior $\rho$, that is,

$$G(\rho) = \mathbb{E}_\rho[L(\boldsymbol{\theta})] = \mathbb{E}_\rho[\mathbb{E}_\nu[-\ln[p(\boldsymbol{y}|\boldsymbol{x}, \boldsymbol{\theta})]]].$$

In fact, Jensen's inequality confirms that the expected Gibbs loss serves as an upper bound for the Bayes loss, i.e., $G(\rho) \geq B(\rho)$. This property supports the expected Gibbs loss to act as a proxy of the Bayes loss, which justifies its usage in gaining insights into how DA impacts the CPE.

We will now study whether data augmentation can cause a CPE on the Gibb loss. In other words, we will examine whether increasing the parameter $\lambda$ of the DA-tempered posterior leads to a reduction in the Gibbs loss. This can be formalized by extending Definition 1 to the expected Gibbs loss by considering its gradient $\nabla_\lambda G(p^\lambda)$ at $\lambda = 1$, which can be represented as follows:

$$\nabla_\lambda G(p^\lambda)_{|\lambda=1} = -\text{COV}_{p^{\lambda=1}}\left(n\hat{L}(D, \boldsymbol{\theta}), L(\boldsymbol{\theta})\right). \tag{9}$$

Where $\text{COV}(X, Y)$ denotes the covariance of $X$ and $Y$. Again, due to the page limit, we postpone the necessary proofs in this section to Appendix B.

With this extended definition, if Eq. (9) is negative, we can infer the presence of CPE for the Gibbs loss as well. Based on this, we say that DA induces a stronger CPE if the gradient of the expected Gibbs loss for the DA-tempered posterior exhibits a more negative trend at $\lambda = 1$, i.e., if $\nabla_\lambda G(p_{\text{DA}}^\lambda)_{|\lambda=1} < \nabla_\lambda G(p^\lambda)_{|\lambda=1}$. This condition can be equivalently stated as

$$\text{COV}_{p_{\text{DA}}^{\lambda=1}}\left(n\hat{L}_{\text{DA}}(D, \boldsymbol{\theta}), L(\boldsymbol{\theta})\right) > \text{COV}_{p^{\lambda=1}}\left(n\hat{L}(D, \boldsymbol{\theta}), L(\boldsymbol{\theta})\right) > 0. \tag{10}$$

The inequality presented above helps characterize and understand the occurrence of a stronger CPE when using DA. A stronger CPE arises if the expected Gibbs loss of a model $L(\boldsymbol{\theta})$ is more *correlated* with the empirical Gibbs loss of this model on the augmented training dataset $\hat{L}_{\text{DA}}(D, \boldsymbol{\theta})$ than on the non-augmented dataset $\hat{L}(D, \boldsymbol{\theta})$. This observation suggests that, if we empirically observe that the CPE is stronger when using an augmented dataset, the set of transformations $\mathcal{T}$ used to generate the augmented dataset are introducing *valuable information* about the data-generating process.

The plots of the last three columns in Figure 2 clearly illustrate such situations. Figure 2d shows that, compared to Figure 2c, the standard DA, which makes use of the invariances inherent in the data-generating distribution, induces a CPE on the Gibbs loss. Thus, the condition in Eq. (10) holds by definition. On the other hand, Figure 2e uses a fabricated DA, where the same permutation is applied to the pixels of the images in the training dataset, which destroys low-level features present in the data-generating distribution. In this case, the gradient of the Gibb loss is positive, and Eq. (10) holds in the opposite direction. These findings align perfectly with the explanations provided above.

## 5.2 DATA AUGMENTATION AND CPE ON THE BAYES LOSS

Now, we step aside of the Gibbs loss and focus back to the Bayes loss. The gradient of the Bayes loss at $\lambda = 1$ can also be written as,

$$\nabla_\lambda B(p^\lambda)_{|\lambda=1} = -\text{COV}_{p^{\lambda=1}}\left(n\hat{L}(D, \boldsymbol{\theta}), S_{p^{\lambda=1}}(\boldsymbol{\theta})\right), \tag{11}$$

where for any posterior $\rho$, $S_\rho(\boldsymbol{\theta})$ is a (negative) performance measure defined as

$$S_\rho(\boldsymbol{\theta}) = -\mathbb{E}_\nu\left[\frac{p(\boldsymbol{y}|\boldsymbol{x}, \boldsymbol{\theta})}{\mathbb{E}_\rho[p(\boldsymbol{y}|\boldsymbol{x}, \boldsymbol{\theta})]}\right]. \tag{12}$$

This function measures the relative performance of a model parameterized by $\boldsymbol{\theta}$ compared to the average performance of the models weighted by $\rho$. Such measure is conducted on samples from the data-generating distribution $\nu(\boldsymbol{y}, \boldsymbol{x})$. Specifically, if the model $\boldsymbol{\theta}$ outperforms the average, we have $S_\rho(\boldsymbol{\theta}) < -1$, and if the model performs worse than the average, we have $S_p(\boldsymbol{\theta}) > -1$ (i.e., the lower the better). The derivations of the above equations are given in Appendix B.

According to Definition 1 and Eq. (11), DA will induce a stronger CPE if and only if the following condition is satisfied:

$$\mathrm{COV}_{p_{\mathrm{DA}}^{\lambda=1}}\left(n\hat{L}_{\mathrm{DA}}(D, \boldsymbol{\theta}), S_{p_{\mathrm{DA}}^{\lambda=1}}(\boldsymbol{\theta})\right) > \mathrm{COV}_{p^{\lambda=1}}\left(n\hat{L}(D, \boldsymbol{\theta}), S_{p^{\lambda=1}}(\boldsymbol{\theta})\right). \tag{13}$$

The previous analysis on the Gibbs loss remains applicable in this context, with the use of $S_\rho(\boldsymbol{\theta})$ as a metric for the expected performance on the true data-generating distribution instead of $L(\boldsymbol{\theta})$. While these metrics are slightly different, it is reasonable to assume that the same arguments we presented to explain the CPE under data augmentation for the Gibbs loss also apply here. The theoretical analysis aligns with the behavior of the Bayes loss as depicted in Figure 2.

Finally, comparing Figure 2d with Figure 3d, we also notice that using a larger neural network enables us to mitigate the CPE because we reduce the underfitting introduced by DA.

**Related work of the data augmentation argument.** The relation between data augmentation and CPE is an active topic of discussion (Wenzel et al., 2020; Izmailov et al., 2021; Noci et al., 2021; Nabarro et al., 2022). Some studies suggest that CPE is an artifact of DA because turning off data augmentation is enough to eliminate the CPE (Izmailov et al., 2021; Fortuin et al., 2022). Our study shows that this is *much more* than an artifact, as also argued in Nabarro et al. (2022). As discussed, the (pseudo) log-likelihood induced by standard DA is a better proxy of the expected log-loss, in the precise sense given by Eq. (10) and Eq. (13).

Other works argue that, when using DA, we are not using a proper likelihood function (Izmailov et al., 2021), and that could be problem. Recent works (Nabarro et al., 2022) have developed principle likelihood functions that integrate DA-based approaches, hoping that this will remove CPE. But they find that CPE still persist. As the methodology described here applies to any likelihood function, we can say that in the experiments of Nabarro et al. (2022) (Figure 4) are also underfitting, and Eq. (13) is satisfied. In fact, probably Eq. (10) is satisfied too.

Another widely accepted viewpoint regarding the interplay between the CPE and DA is that DA increases the effective sample size (Izmailov et al., 2021; Noci et al., 2021), "intuitively, data augmentation increases the amount of data observed by the model, and should lead to higher posterior contraction" (Izmailov et al., 2021). In that sense, our study aligns with this interpretation because CPE needs higher $\lambda$ values to alleviate underfitting, which, in turn, leads to higher posterior concentration. But, our analysis is more nuanced because we show that DA only induces CPE if it provides meaningful information about the data-generating process. Thus, higher posterior concentration in the context of non-meaningful DA does not improve performance; as discussed before, Figure 2e illustrates this situation.

## 6  CONCLUSIONS

In this work, we show that the presence of CPE implies that the Bayesian posterior is underfitting. As discussed in previous works, CPE also implies that that the likelihood and/or the prior are misspecified. In consequence, it is reasonable to deduce that CPE is a problem of underfitting induced by poorly specified likelihoods (e.g., softmax-based likelihoods on curated datasets) and/or priors (e.g., overregularizing priors).

This insight provides a practical recommendation for addressing CPE: to use more powerful likelihood functions (e.g., by using larger models) and better specified priors that do not overregularize. This seems to be specially relevant when using DA because, as discussed above, it induces a stronger CPE because the augmented data contains more information about the data-generating distribution and, in consequence, it is *safer to fit it further*. At the same time, tuning the hyper-parameter $\lambda$ is also a valid strategy to improve performance but, also, to detect if the Bayesian posterior underfits.

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
