# OpenReview forum: "If there is no underfitting, there is no Cold Posterior Effect"
_ICLR.cc/2024/Conference — Submitted to ICLR 2024_

### Official Review · Reviewer_B5C9 · 2023-10-31

**Soundness:** 2 fair
**Presentation:** 3 good
**Contribution:** 3 good
**Rating:** 5
**Confidence:** 4

**Summary:**

This work investigates the so-called cold posterior effect, a phenomenon in Bayesian deep learning where it is observed that tempering the Bayesian posterior can counter-intuitively lead to improved test performance. The authors show how the CPE arises as a consequence of models under-fitting the data and demonstrate how this can be remedied precisely by lowering the temperature. They connect their findings with previously established theories surrounding the CPE and show that most such explanations can be unified within their framework.

**Strengths:**

1. The problem of the cold posterior effect has lead to a plethora of works aiming at understanding it and most of these explanations tackle very different aspects of this problem in very different ways. I think this work is a timely contribution as it tries to unify all these previous approaches in a single more “global” lens. The authors also show great knowledge of the related works, covering most of the contributions to this problem.
2. I really find the result in Theorem 4 quite interesting and also somewhat surprising. I think this is actually the core result of this work although it is not really highlighted like that.

**Weaknesses:**

1. I find the claims in this work somewhat misleading. The main contribution is the fact that if the posterior is not underfitting, then there is no cold posterior effect. But this statement is almost trivial as (almost) all that is to prove is already assumed in the statement. Underfitting, as defined by the authors, means “a model having training and testing losses much higher than they could be i.e. there exists another model in the model class having simultaneously lower training and testing losses.” Taking the opposite, no underfitting basically means that there is no model with better train and test loss, hence if the Bayesian posterior is not underfitting then of course there is no cold posterior effect. Am I missing something here?  The authors then show that posteriors with smaller temperature having smaller training loss, which is not surprising since the effect of the prior is essentially down-weighted and the model concentrates more on the likelihood, which exactly guarantees a better fit. This, not very surprising fact does all the heavy-lifting here: If one now assumes CPE, then yes, there is a model with smaller temperature and better test loss, which also has smaller training loss but again, not much has been shown here. I also find all the mathematical notation more confusing than helpful here, it makes this relatively simple fact seem more complicated than it really is. I think my issues are somewhat addressed by Theorem 4 which seems to more strongly show the claims made **before** presenting it. I would really appreciate if the authors could clear up my confusion as to what is assumed when words like “underfitting” are used, and what is proved rigorously.
2. Theorem 4 is a very interesting result and I would have loved to see some empirical verification of it, i.e. if the posterior is optimal at $\lambda = 1$, does the train loss really not change anymore if you add a new sample to the posterior (of course one would need to average over adding the sample)? Can you basically predict optimality of $\lambda=1$ from this property alone?
3. I think one important theoretical work in the literature regarding the origins of CPE in conjunction with data augmentation has not been treated in this work. While you cite [1] for re-scaling the prior, you don’t discuss their results regarding data augmentation and how the correlations of errors, arising from assigning the same label to augmented samples, influences the resulting posterior. It is not obvious to me at first glance how those results can be casted within the under-fitting hypothesis of this work, especially the correlated nature of the data. Could you elaborate on the connection to your work?


[1] Bachmann et al., How Tempering Fixes Data Augmentation in Bayesian Neural Networks

**Questions:**

In Proposition 3, how does the underlying data distribution $\nu$ enter the picture here? The CPE is clearly a function of the data distribution $\nu$ as it talks about generalisation loss, but the inequality seems to be independent of $\nu$ except for involving training samples that are drawn from it. Could you elaborate how you are able to make this connection?

---

> ### Author Response · Authors · 2023-11-16
>
> Thanks for your feedback. Let us try to clarify the confusion.
>
> > Taking the opposite, no underfitting basically means that there is no model with better train and test loss, hence if the Bayesian posterior is not underfitting then of course there is no cold posterior effect. Am I missing something here?
>
>
>
> This kind of reasoning can be tricky, we will try to explain here why “no underfitting implies no CPE” cannot be directly proved by simply inverting the definition (without using Theorem 2).
>
> It’s correct that no underfitting means there is no model with **both** better training **and** better testing loss **simultaneously**. Hence, the following must hold if there is no underfitting:
> 1. If another model has better train loss, it must have worse test loss.
> 2. If another model has better test loss, it must have worse train loss.
>
> The situation, where increasing $\lambda>1$ gives a better testing loss (there is CPE), itself contradicts neither situation 1) nor 2). It is because it could be the case where the tempered posterior has worse train loss (situation 2). Therefore, no underfitting and CPE could co-exist.
>
> However, situation 2) cannot happen as Theorem 2 shows that the train loss is also enhanced by increasing $\lambda>1$. If you do not use Theorem 2 and try to show the implication from the definition, you do not reach any contradiction.
>
> > In Proposition 3, how does the underlying data distribution enter the picture here?
>
> Proposition 3 is a consequence of Proposition 6 in Appendix A.3 and the definition of CPE. To shed some light here, Proposition 6 says essentially that the gradient of the Bayes loss has an equivalent form as the difference between the Gibbs loss of the updated posterior (Eq. (6)) and the tempered posterior; where the “data-generating distribution” appears in the updated posterior. Next, the definition of CPE says the gradient of the Bayes loss is negative $< 0$, i.e, the Gibbs loss of the updated posterior is lower than the Gibbs loss of the tempered posterior. Hence, the Gibbs loss of the posterior cannot reach the minimum, which is the message of Proposition 3. Therefore, we don’t see the data-generating distribution in the conclusion of Proposition 3 since the related quantity is thrown away in some sense in the middle of derivation.
>
> >Theorem 4 is a very interesting result and I would have loved to see some empirical verification of it, i.e. if the posterior is optimal at lambda=1, does the train loss really not change anymore if you add a new sample to the posterior (of course one would need to average over adding the sample)? Can you basically predict optimality of lambda=1 from this property alone?
>
> We are really glad that you can appreciate the relevance of this result. Yes, if $\lambda=1$ is optimal, then the training loss does not change for the updated posterior (in expectation). Please note Thm 4 establishes a  “if and only if” condition.
>
> We will try to highlight this result more in the new version. And we also plan to include a table like the next one illustrating this result for the linear regression model of Figure 1 under the four setups depicted there.
>
> |               | Fig.1(a): no CPE | Fig.1(b): WPE | Fig.1(c): CPE-likelihood | Fig.1(d): CPE-prior |
> |--------------------------------------------|------------------|---------------|--------------------------|---------------------|
> | $\hat{G}(p,D)$                             | 1.521            | 0.467         | 1.922                    | 1.996               |
> | Mean of 10 estimates of $\hat{G}(\bar{p},D)$                       | 1.519            | 0.536         | 1.888                    | 1.905               |
> | Standard Deviation of 10 estimates of $\hat{G}(\hat{p},D)$ | 0.009            | 0.013         | 0.009                    | 0.009               |
>
> As can be seen, when there is no CPE the train loss does not change, as predicted by Thm 4. And, as Proposition 6 in the Appendix also predicts, when there is a “warm posterior effect”, the train loss of the expected updated posterior becomes bigger.  And when there is CPE, the train loss of the expected updated posterior becomes smaller. We plan to add this discussion to the main paper. Thanks for bringing this up!

---

> > ### Author Response · Authors · 2023-11-16
> >
> > > I think one important theoretical work in the literature regarding the origins of CPE in conjunction with data augmentation has not been treated in this work. While you cite [1] for re-scaling the prior, you don’t discuss their results regarding data augmentation and how the correlations of errors, arising from assigning the same label to augmented samples, influences the resulting posterior. It is not obvious to me at first glance how those results can be casted within the under-fitting hypothesis of this work, especially the correlated nature of the data. Could you elaborate on the connection to your work?
> >
> > We also thank the reviewer for bringing the connection to this paper up. We have given new thoughts to this work and we have realized our work aligns with the findings of [1] in relation to the degree of model invariances with respect to the data augmentation. For example, if the model is completely invariant to the data augmentation, we would have that $\hat{L}_{DA}(D,\theta)=\hat{L}(D,\theta)$.. And, in consequence, $p^{\lambda=1}_DA=p^{\lambda=1}$  and the inequality of Eq. (10) will be an equality. I.e., DA would not cause a stronger CPE. If there is no CPE under the standard Bayesian posterior (i.e, $\lambda=1$ is optimal), using DA over a completely invariant model would not induce any CPE (i.e., $\lambda=1$  would be optimal as well). We plan to elaborate on this and include this connection in the final version of the paper.

---

> > > ### Comment · Reviewer_B5C9 · 2023-11-20
> > > **Reply**
> > >
> > > I thank the authors for their response!
> > >
> > > **No underfitting --> No CPE:** I'm still not convinced by this point and especially the "amount" of that statement that the authors actually need to prove.
> > >
> > > Let's assume that we have no underfitting of the posterior, i.e. the posterior $f$ is optimal and any other (tempered) posterior $f^T$ that reaches better training loss must have worse test loss (this all comes from the definition of overfitting). Now it is not very surprising that reducing the temperature leads to better training loss as we upscale the importance of the likelihood, reducing the regularisation effect of the prior.  If one accepts that Bayesian models tend to fit their training data slightly worse than the same models purely optimised in an MLE-sense (which is pretty much known to any ML practitioner), then the claim basically becomes a consequence of the definition. That's why I feel like the definition of "underfitting" basically does all the heavy-lifting. Theorem 2 makes the "T small --> training loss small" mathematically explicit but also makes things look more complicated than they really are. All that is needed for the core-characterisation is that Bayesian models fit worse on training data, so to me the insights from this must be limited.
> > >
> > > **Experiments for Thm 4:** Thanks for performing these experiments, that's a nice result.
> > >
> > > **Comparison to [1]**: Thank you for this comparison, I think this makes sense. I guess the fact that samples are correlated can be absorbed into the underfitting notion, which does not care about how samples correlate? In a sense that is nice as it makes things easier, but again I have the feeling that such a point of view might miss important details. Or how would the fact that observations (or more precisely, errors) are correlated enter into your framework?
> > >
> > > Overall, I remain reluctant to increase my score as the entire paper builds upon the notion of underfitting, which again to me is almost a rephrasing of the cold posterior effect as it only relies on the fact that Bayesian models fit their training data worse. I might still be missing something on the other hand and I ask the authors to point out if something in my assessment is wrong.

---

> > > > ### Author Response · Authors · 2023-11-21
> > > >
> > > > Thank you for the quick and concrete reply. First, Let us focus on the main question: does this paper merit acceptance? The two main criteria are novelty and relevance to the community. The
> > > >
> > > > - **Novelty**. Let us stress this again. Despite being simple, straightforward, and intuitive, the connection between CPE and underfitting is not present in the literature. We have not found any reference. Only a few papers mentioned that it was traditionally known that Bayesian neural networks are known to underfit. But the link to CPE is non-existent in the literature. **Our paper should be judged by the novelty of the contribution, not by the complexity of the contribution.**
> > > >
> > > >
> > > > - **Relevance**. That CPE is a sign of underfitting is relevant, despite being simple to deduce. The community has mostly focused so far on CPE as a likelihood/prior misspecification or data augmentation problem. Being fully aware that this is also an underfitting problem is very relevant. Please note that misspecification and DA do not necessarily imply underfitting (Fig.1b shows how misspecification can also lead to overfitting; Fig.2e shows how DA can also lead to overfitting). **Our paper should be judged by the relevance to the community of the existence of this connection between CPE and underfitting, not by the complexity of the logical reasoning involved in arriving at this connection.**
> > > >
> > > >
> > > > Finally, the presented mathematical formalization of CPE mainly aims to make the connection between CPE and underfitting formally correct. The mathematical formalization itself is simple and, in exchange for certain extra complexity, provides some non-trivial insights like Thm 4 and the connection with data correlation in DA. In particular, the link between data correlation in DA with CPE comes from the equivalent formula of the definition of CPE (Proposition 6), which makes the characterization explicit, but intuitively simple. The amount of correlation will suggest the strength of the CPE. The alignment with our characterization of CPE and underfitting exactly emphasizes the explainability of our simple connection. Even so, we plan to give, at the very beginning of Section 3,  the intuitive reasons for this connection.

---

### Official Review · Reviewer_MuTU · 2023-10-31

**Soundness:** 2 fair
**Presentation:** 3 good
**Contribution:** 3 good
**Rating:** 5
**Confidence:** 3

**Summary:**

The paper presents in a theoretical and empirical setup that the presence of CPE implies the existence of under-fitting along with previous evidence of CPE such as misspecification of the prior and the likelihood. That is in a way that the misspecification of the prior or the likelihood have underfitting as an outcome and therefore the CPE.

**Strengths:**

-Interesting take on the CPE problem that might give light to new avenues on why CPE exists and how to tackle it.
-Potential of good value if the argument is made more clear or presented in a better way.

**Weaknesses:**

-The argument is a bit unclear from my perspective. The authors argue that the problem is under-fitting which comes from misspecification. So is the problem the under-fitting or the misspecification which causes the under-fitting.
-The paper seems to be stepping on previous results and works, claiming that the misspecification of the prior or the likelihood lead to underfitting, and therefore underfitting is the problem that causes CPE. Well if CPE is present when under-fitting is present then the problem falls on either the prior or the likelihood.
-Some of the results are not convincing. A simple linear regression model on synthetic data is not enough in Figure 1. Figure 2 and 3 are also a bit unclear.

**Questions:**

My main question is what is the argument that the authors are making? Reading the paper it looks like you are arguing that misspecification is causing the CPE by causing underfitting. It is well known that any misspecification in Bayesian setups leads to lower performance.

I think I have understood the math and the experiments of the paper but the main argument does not seem very novel. Can the authors better explain?

One tip that I can give on the presentation is to add the results of the big models in the main text and the toy experiments in the Supplement. Or have a gradual increase of difficulty in the presentation of the experiments. For instance you start with linear regression on synthetic data, then you have a ResNet experiment on CIFAR10 and then on Imagenet.

---

> ### Author Response · Authors · 2023-11-16
>
> Thanks for your feedback. Let us try to clarify the confusion.
>
> > The argument is a bit unclear from my perspective. The authors argue that the problem is under-fitting which comes from misspecification. So is the problem the under-fitting or the misspecification which causes the under-fitting
>
> > The paper seems to be stepping on previous results and works, claiming that the misspecification of the prior or the likelihood lead to underfitting, and therefore underfitting is the problem that causes CPE. Well if CPE is present when under-fitting is present then the problem falls on either the prior or the likelihood.
>
> > My main question is what is the argument that the authors are making? Reading the paper it looks like you are arguing that misspecification is causing the CPE by causing underfitting. It is well known that any misspecification in Bayesian setups leads to lower performance.
>
> > I think I have understood the math and the experiments of the paper but the main argument does not seem very novel. Can the authors better explain?
>
> To start, we would like to clarify that the main takeaway message of our work is the following: **The presence of CPE implies that our Bayesian setup is misspecified and underfitting at the same time.**
>
> This is a simple, clean but actually **novel** way to interpret CPE.
> 1. Connections to previous work: The connection with underfitting has not been made before when trying to understand CPE. Most of the current literature focuses on the misspecification problem itself, but **not on misspecification in the context of underfitting**. We find it quite important that the ML community is fully aware of this point.
> 2. “it is well known that any misspecification in Bayesian setups leads to lower performance.”: We totally agree. But this “lower performance” can be due to **underfitting or overfitting**, i.e., you can have a misspecified model that is severely overfitting your training data, as we show in Figure 1 (b). Understanding which of the two is present when we observe CPE is quite relevant. Reviewer wHkC states that this question “ hold[s] profound implications.”
>
> Having said that, we agree that we could do better in characterizing the relationship between misspecification and underfitting and will do so in the final version.
>
> >Some of the results are not convincing. A simple linear regression model on synthetic data is not enough in Figure 1. Figure 2 and 3 are also a bit unclear.
>
> >One tip that I can give on the presentation is to add the results of the big models in the main text and the toy experiments in the Supplement. Or have a gradual increase of difficulty in the presentation of the experiments. For instance you start with linear regression on synthetic data, then you have a ResNet experiment on CIFAR10 and then on Imagenet.
>
> Thanks for this tip! We will consider your advice in the next version.

---

> > ### Comment · Reviewer_MuTU · 2023-11-22
> >
> > I thank the authors for the response. After reading Reviewer B5C9's review and discussion, I understand that we have a similar way of seeing the situation. I am still a bit reluctant about increasing my score, since the argument made here is not novel. I will keep my score of 5. I would suggest the authors to rethink of the way of presenting the argument as something different than underfitting.

---

> > > ### Author Response · Authors · 2023-11-22
> > >
> > > Thank you for your reply. But, honestly, we will love to find previous work where the connection between CPE and underfitting is discussed.  We have not found any reference. Only a few papers mentioned that it was traditionally known that Bayesian neural networks are known to underfit. But the link between  CPE and underfitting is non-existent in the literature.
> > >
> > > Please note that misspecification and DA do not necessarily imply underfitting and CPE (Fig.1b shows how misspecification can also lead to overfitting and WPE; Fig.2e shows how DA can also lead to overfitting and WPE;  (Domingos, 2000) also discusses this).
> > >
> > > Our paper should be judged by the novelty of the contribution, not by the complexity of the contribution,  which, **in hindsight**, looks simple and straightforward.

---

### Official Review · Reviewer_wHkC · 2023-11-01

**Soundness:** 2 fair
**Presentation:** 3 good
**Contribution:** 2 fair
**Rating:** 5
**Confidence:** 4

**Summary:**

- The paper claim that if there is cold posterior effect, it means the posterior distribution is underfitted.

- They provide theoretical and experimental evidence which support their claim.

**Strengths:**

- They address a highly significant issue, and their arguments hold profound implications. If properly substantiated, their work is poised to be regarded as a pivotal study.

- Their notation is clean, and well-written.

**Weaknesses:**

$\bf{(Major)}$

According to the definition of CPE, as lambda increases, the test loss should decrease. Additionally, as $\lambda$ increases, the train loss always decreases irrespective of CPE. Up to this point, arguments in Theorem 2, Proposition 3 are accurate but rather self-evident.

However, they have not demonstrated that the distribution, whose existence was proven, becomes a posterior distribution (which should be definable from a new likelihood or a new prior). Hence, according to the definition “underfitting” defined by authors, Insight 1 may not be correct.

After demonstrating that, I believe at least one of the following two pieces of evidence should be provided.
1) The original likelihood (or prior) is misspecified, compare to new likelihood (or prior).
2) An inverse proposition also holds.(i.e., If there is underfitting, there exists CPE).

In the current manuscript, I believe that their main claim has not been theoretically clarified.

$\bf{(Minor)}$

It seems that Figure 2 and Figure 3 do not connect well with the main claims of the paper. The common implication in the results appears to merely suggest that "CPE exists in image data analysis."

**Questions:**

In page 3, authors state that

“In the context of Bayesian inference, we argue that the Bayesian posterior is underfitting if there exists another posterior distribution with lower empirical Gibbs and Bayes losses at the same time.”

Is this a commonly accepted concept or a newly proposed one? If it's the former, it seems appropriate to provide references.

---

> ### Author Response · Authors · 2023-11-16
>
> Thanks for your feedback. Let us try to clarify in the following.
>
> > According to the definition of CPE, as $\lambda$ increases, the test loss should decrease. Additionally, as  $\lambda$  increases, the train loss always decreases irrespective of CPE. Up to this point, arguments in Theorem 2, Proposition 3 are accurate but rather self-evident.
>
> As we have answered in the overall rebuttal at the top of the page, we do agree that the connection is, in some sense, straightforward.  However, it's noteworthy that this link has not been recognized in the literature. The primary contribution of our study is to make the community aware of this connection. Because it is essential for comprehending and addressing the challenges associated with CPE.
>
> Having said that, our formalization also provides new insights into CPE. Theorem 4 provides an interesting characterization not done before. Reviewer B5C9 is very positive about it. In our opinion, the connection with Data Augmentation does also provide new insights.
>
>
>
> > However, they have not demonstrated that the distribution, whose existence was proven, becomes a posterior distribution (which should be definable from a new likelihood or a new prior). Hence, according to the definition “underfitting” defined by authors, Insight 1 may not be correct.
>
>
> Let us first clarify that, in our work, we refer to a “posterior distribution” to **any distribution over the parameter space which depends on the training data**. We refer to the “Bayesian posterior distribution” to a “posterior distribution” which is the result of the strict application of the Bayes’ rule. I.e.,  $p^\lambda(\theta|D)$ is a posterior distribution, while $p^{\lambda=1}(\theta|D)$ is the Bayesian posterior distribution. We will make this distinction crystal clear in our paper.
>
> Therefore, Insight 1 is correct as we said “there exists **another posterior**, rather than **another Bayesian posterior**, that has lower training and testing loss at the same time.”
>
> As you say,  it would be great to know if this tempered posterior can be attained by some other Bayesian posterior coming from another likelihood/prior. We think this indeed is an interesting question. Although we think it is out of the scope to include this in our paper, where the focus is to highlight the connection between CPE and underfitting that has been overlooked so far, there are references showing positive answers, which we provide in the next section.
>
> >  I believe at least one of the following two pieces of evidence should be provided.
> The original likelihood (or prior) is misspecified, compared to new likelihood (or prior).
> An inverse proposition also holds.(i.e., If there is underfitting, there exists CPE).
>
> The first point is highly related to the previous question; it is clear that if CPE is present, prior and/or likelihood are misspecified, otherwise the Bayesian posterior would be optimal and CPE would not be present. As far as we know, at least in certain cases, the tempered posterior can be derived as the Bayesian posterior with a proper likelihood and prior (Wenzel et al., 2020 Wilson, Izmailov, 2020, Nabarro et al., 2022). Please refer to those for more details.
>
>
> Regarding the inverse proposition; whether it holds is also a good question. However, it is not true in general; consider the extreme case where the prior distribution is a delta distribution centered at a model that does not explain either train or test data, let’s say $\theta_0$. In this setting, the Bayesian posterior is
> $$
> p(\theta|D) \propto P(D|\theta)p(\theta),
> $$
> which is also a delta distribution centered at $\theta_0$. The same happens with any tempered posterior. Even if the model $\theta_0$ is underfitting (there exist models with better training and testing loss), CPE does not appear in this context.
>
> > In the current manuscript, I believe that their main claim has not been theoretically clarified.
>
> With the above clarifications, our claim “If CPE is present, there exist alternative **posteriors** with better train and test loss” theoretically holds. As said before, what we don’t show is that “If CPE is present, there exist alternative **Bayesian posteriors** with better train and test loss” theoretically holds. We are fully aware this would be a stronger result. But we do consider this paper as a first step in this direction. This work can help to bring attention to this relevant open questions.

---

> > ### Author Response · Authors · 2023-11-16
> >
> > > It seems that Figure 2 and Figure 3 do not connect well with the main claims of the paper. The common implication in the results appears to merely suggest that "CPE exists in image data analysis."
> >
> > We will try to summarize the figures here and will update them in the final version of the paper.
> >
> > All plots in the figures are used to support our main claim: **if we mitigate underfitting, which can be due to several different reasons (below), we will observe less CPE**. In particular, we demonstrate:
> > 1. The effect of the priors in Figures 2a compared to 2c (as well as 3a compared to 3c): When a sharp prior underfits the data, using wider priors reduces the underfitting, and thus has less CPE.
> > 2. The effect of likelihood misspecification in Figures 2b compared to 2c (as well as 3b compared to 3c): When a softmax likelihood underfits the data (a curated dataset in our example), using shaper (less misspecified in this context) likelihoods reduces underfitting, thus has less CPE.
> > 3. The effect of informative data augmentation in Figure 2d compared to 2c (as well as 3d compared to 3c): When removing DA, there is less information in the dataset, making the model easier to fit the data, i.e., less underfitting, and thus has a weaker CPE.
> > 4. The effect of the model size (compare Figure 2 to Figure 3): When the models are bigger and more capable, it’s easier for the model to fit the data, thus there will be less underfitting and less CPE.
> >
> > > In the context of Bayesian inference, we argue that the Bayesian posterior is underfitting if there exists another posterior distribution with lower empirical Gibbs and Bayes losses at the same time.”Is this a commonly accepted concept or a newly proposed one? If it's the former, it seems appropriate to provide references.
> >
> > We tried to find definitions of underfitting in the context of Bayesian methods, but to the best of our knowledge, we were not able to find a commonly adopted definition. The definition we use here aligns with what is intuitively understood as underfitting in non-Bayesian settings.

---

> > > ### Comment · Reviewer_wHkC · 2023-11-23
> > >
> > > Thank you for authors’ responses and discussions.
> > > The rebuttal has helped me understand exactly what the paper's claims are and what implications it intends to give.
> > > Their claim (Insight 1) is valid, and I understand that it is logically flawless.
> > > However, after hearing their definition of 'posterior distribution,' my impression that it resembles an obvious proposition akin to wordplay has intensified.
> > > Additionally, even if this state has not existed before, I am especially doubtful about what implication this state can provide for AI.
> > > For these reasons, I have decided not to lower or raise the score.

---

> > > > ### Author Response · Authors · 2023-11-23
> > > >
> > > > Dear reviewer,
> > > >
> > > > Thank you for the response. However, as you mentioned before this work "address a highly significant issue, and their arguments hold profound implications. If properly substantiated, their work is poised to be regarded as a pivotal study.” we are curious to hear what exactly made you change your mind.
> > > >
> > > > Our work is valuable because
> > > > 1. Nowadays, with overparameterized models, **the problem of overfitting is more profound in most of the domains**. In particular, there is a tutorial at NeurIPS this year on “Reconsidering Overfitting in the Age of Overparameterized Models.” However, the finding of CPE in Bayesian, on the other hand, points to underfitting. It reminds the Bayesian community to examine this clear gap.
> > > > 2. It also reminds the community that **before breaking heads to come up with complicated remedies for CPE, we should first think about whether there is clear underfitting, which might open the way to some easy solutions to address the underfitting. For example, increasing model capacity or using alternative priors that do not overregularise.**

---

### Official Review · Reviewer_dcZF · 2023-11-08

**Soundness:** 4 excellent
**Presentation:** 3 good
**Contribution:** 4 excellent
**Rating:** 8
**Confidence:** 3

**Summary:**

The authors investigate the cause of the cold posterior effect (CPE) and argue that it is due to model underfitting. They do this both analytically and empirically; they show theoretically that the CPE implies underfitting and further construct experiments to analyse the influence of likelihood and prior misspecification as well as data augmentation. They conclude that each of these settings induces a CPE only if they cause the posterior to underfit. They further show that the CPE exists even for models that allow for exact Bayesian inference and that a "warm" posterior effect can occur too when the likelihood is misspecified. Together, these results explain why certain priors and likelihoods can lead to a CPE, as well as why larger models suffer less from the CPE.

**Strengths:**

1. The paper presents an elegant unification of previous explanations and observations of the CPE as being due to underfitting. It explains both the role of prior or likelihood misspecification as well as addresses the observed CPE when using data augmentations, which is really remarkable.
2. The experiments are cleverly designed and convey some great insights into the CPE.
3. The paper is well-written, and the authors do a great job of explaining ideas and results that are quite abstract.
4. The paper is original and of high technical quality. It seems bound to become extremely significant for understanding the CPE and will therefore be of great interest to the ICLR community.

**Weaknesses:**

This might be the first time I have struggled to find weaknesses in a paper. While the usual do-more-experiments comment can always be used, I cannot think of particular experiments that would dramatically contribute to the paper. It seems to be all-round solid work.

**Questions:**

**Questions**
1. Your results indicate that larger model capacities should be less susceptible to the CPE since they are less likely to underfit. However, Wenzel et al. (2020), figure 11, actually show the opposite, namely that increasing the depth of an MLP has no effect on the CPE, and that increasing the width makes the effect more pronounced. Do you have any thoughts on this?
2. I am trying to understand what the next steps for research on this topic might be. Did you observe any situations where underfitting could not explain the CPE?
3. Do you have any suggestions for future work?


**Suggestions**
- Minor thing: the plots seem to have been rasterised with such high resolution that my pdf viewer struggles. I would suggest using vector graphics instead.

---

> ### Author Response · Authors · 2023-11-16
>
> Thanks for your positive review. We answer to your questions below:
>
> >Your results indicate that larger model capacities should be less susceptible to the CPE since they are less likely to underfit. However, Wenzel et al. (2020), figure 11, actually show the opposite, namely that increasing the depth of an MLP has no effect on the CPE, and that increasing the width makes the effect more pronounced. Do you have any thoughts on this?
>
> Thanks a lot for pointing us to this figure. We did not realize this experiment was present there. One of the possible explanations is that they are using full-tempering while we are using likelihood tempering.
> 1. For full-tempering, Theorem 2 does not necessarily hold anymore. Intuitively, since \lambda works on both likelihood (data) and prior (regularization) together, the effect of increasing \lambda is mixed, thus not necessarily improving the fit on training data.
> 2. Thus, the CPE brought by full-tempering, as \lambda increases, does not necessarily come with a better training loss, i.e., training loss may not be improvable. Hence, full-tempered CPE can no longer be interpreted as just underfitting only.
> 3. As such, for full-tempering, increasing the model capacity may not achieve a lower degree of CPE as in our case.
> However, we believe it’s still valuable and will reproduce this experiment with likelihood-tempering to clear the argument.
>
> > I am trying to understand what the next steps for research on this topic might be. Did you observe any situations where underfitting could not explain the CPE?
>
> If there is CPE, then your Bayesian posterior is underfitting. But, it might be the case that the posterior is underfitting and you do not have CPE. We haven’t said anything about the reverse implication. But, we do think this implication does not really hold in general. In the response to Reviewer wHkC we provide a counterexample.
>
> For us, a relevant question is: is there any problem in working with tempered posteriors? What is preventing us from tuning this hyperparameter when using a Bayesian approach? This lambda hyper-parameter could be tuned using, for example, an independent validation dataset. Providing theoretical arguments in favor of this approach would be valuable. This work would support this, because if we do not do it, we know we are not squeezing out all the power of (generalized) Bayesian methods.

---

### Author Response · Authors · 2023-11-16
**General response**

We sincerely thank the reviewers for their positive feedback and also their critics. Let us quickly summarize the main takeaways of our paper together with the main positive and negative points raised by the reviewers.

Most of the previous works studying CPE depart from the following statement (Gelman et al., 2013):

If the prior and the likelihood are well specified then Bayesian posterior is optimal. In consequence, the presence of the CPE shows that either the prior, the likelihood or both are misspecified.

Our point is that the above implication does not provide an accurate understanding of CPE.
Reviewer MuTU mentions that “It is well known that any misspecification in Bayesian setups leads to lower performance”. Yes, this is true. But this lower performance can be due to an underfitting or to an overfitting problem. This has been widely discussed in the literature (Domingos, 2000; Immer et al., 2021; Kapoor et al.,2022). We also illustrate this in Figure 1 (b).
Understanding which of the two is present when we observe CPE is quite relevant. If you know your misspecified likelihood/prior are underfitting, you would change them in different ways if you know they are overfitting.
Reviewer wHkC states that this question “ hold[s] profound implications. If properly substantiated, their work is poised to be regarded as a pivotal study.”

In this work, we simply aim to send the message that CPE implies underfitting.
Our work provides a mathematical formalization (of many possible) of this connection.

A main concern of several reviewers is that this connection is “self-evident”
We do agree that the connection is, in some sense, straightforward.
But, surprisingly, it is not acknowledged in the literature. For example, a simple web search of the terms “cold posterior effect” and “underfitting” hardly provides references simultaneously containing both terms.

The main contribution of this work is to highlight this connection.
We think it is quite relevant that Bayesian ML community is aware of this connection, because it will help to better understand and alleviate CPE.

---

> ### Author Response · Authors · 2023-11-22
> **Novelty and Relevance of Our Work**
>
> We thank the reviewers for the interactions and communications. During the discussions, however, we realized there was a fundamental misunderstanding in interpreting our work, which led to concerns about novelty. With respect to that, we would like to clarify here:
>
> 1. **Novelty**. Let us stress this again. Despite being simple, straightforward and intuitive, the connection of CPE and underfitting is not present in the literature. We have not found any reference. Only a few papers mentioned that it was traditionally known that Bayesian neural networks are known to underfit. But the link between  CPE and underfitting is non-existent in the literature. **Our paper should be judged by the novelty of the contribution, not by the complexity of the contribution**,  which, **in hindsight**, looks simple and straightforward.
>
> 2. **Relevance**. That CPE is a sign of underfitting is relevant, in spite of being simple to deduce. The community has mostly focused so far on CPE as a likelihood/prior misspecification problem. Being fully aware that this is also an underfitting problem is very relevant. Please note that misspecification and DA do not necessarily imply underfitting and CPE (Fig.1b shows how misspecification can also lead to overfitting and WPE; Fig.2e shows how DA can also lead to overfitting and WPE;  (Domingos, 2000) also discusses this).
>
> **Our paper should be judged by the relevance to the community of the existence of this connection between CPE and underfitting, not by the complexity of the logical reasoning involved in arriving at this connection** which, **in hindsight**, looks simple and straightforward.

---

### Meta-Review · Area_Chair_WRoY · 2023-12-14

**Metareview:**

This paper provides a mathematical and empirical study showing that the cold posterior effect, which can arise from model misspecification, in particular is linked to underfitting (rather than overfitting) in deep learning models. While this precise result has not been previously formulated in such a way in analysis of the cold posterior effect, three of the four reviewers were left unconvinced of its practical utility, and took quite some discussions with the reviewers to clarify ways in which this does not follow immediately from the chosen definitions of cold posteriors and underfitting in the context of Bayesian models. Given the degree to which many of the reviewers have been left uncertain, I think the paper should not be published at ICLR at this time, as such difficulties are likely to arise from other members of the community. I would suggest that a revision of the paper focus on clearly articulating the impact of the contribution, as well as formulating the definitions and presentation in such a way as to sidestep the issues encountered by these reviewers; doing so would likely lead to a more impactful paper.

**Justification For Why Not Higher Score:**

The presentation of the argument ultimately led to quite a bit of confusion for the reviewers. This could be resolved in a revised version of the paper, with careful attention to leading the reader through the definitions, arguments, and experiments.

**Justification For Why Not Lower Score:**

N/A

---

### Decision · Program_Chairs · 2024-01-16

Reject